# Scalable Sensitivity and Uncertainty Analyses for Causal-Effect Estimates of Continuous-Valued Interventions

**Andrew Jesson**\*
OATML
Department of Computer Science
University of Oxford

**Alyson Douglas**
AOPP
Department of Physics
University of Oxford

**Peter Manshausen**
AOPP
Department of Physics
University of Oxford

**Maëlys Solal**
Department of Computer Science
University of Oxford

**Nicolai Meinshausen**
Seminar for Statistics
Department of Mathematics
ETH Zurich

**Philip Stier**
AOPP
Department of Physics
University of Oxford

**Yarin Gal**
OATML
Department of Computer Science
University of Oxford

**Uri Shalit**
Machine Learning and Causal Inference in Healthcare Lab
Technion – Israel Institute of Technology

## Abstract

Estimating the effects of continuous-valued interventions from observational data is a critically important task for climate science, healthcare, and economics. Recent work focuses on designing neural network architectures and regularization functions to allow for scalable estimation of average and individual-level dose-response curves from high-dimensional, large-sample data. Such methodologies assume ignorability (observation of all confounding variables) and positivity (observation of all treatment levels for every covariate value describing a set of units), assumptions problematic in the continuous treatment regime. Scalable sensitivity and uncertainty analyses to understand the ignorance induced in causal estimates when these assumptions are relaxed are less studied. Here, we develop a continuous treatment-effect marginal sensitivity model (CMSM) and derive bounds that agree with the observed data and a researcher-defined level of hidden confounding. We introduce a scalable algorithm and uncertainty-aware deep models to derive and estimate these bounds for high-dimensional, large-sample observational data. We work in concert with climate scientists interested in the climatological impacts of human emissions on cloud properties using satellite observations from the past 15 years. This problem is known to be complicated by many unobserved confounders.

## 1 Introduction

Understanding the causal effect of a continuous variable (termed "treatment") on individual units and subgroups is crucial across many fields. In economics, we might like to know the effect of price on demand from different customer demographics. In healthcare, we might like to know the effect of medication dosage on health outcomes for patients of various ages and comorbidities. And in

---

\*Correspondence to `andrew.jesson@cs.ox.ac.uk`

36th Conference on Neural Information Processing Systems (NeurIPS 2022).

climate science, we might like to know the effects of anthropogenic emissions on cloud formation and lifetimes under variable atmospheric conditions. In many cases, these effects must be estimated from observational data as experiments are often costly, unethical, or otherwise impossible to conduct.

Estimating causal effects from observational data can only be done under certain conditions, some of which are not testable from data. The most prominent are the common assumptions that all confounders between treatment and outcome are measured ("no hidden confounders"), and any level of treatment could occur for any observable covariate vector ("positivity"). These assumptions and their possible violations introduce uncertainty when estimating treatment effects. Estimating this uncertainty is crucial for decision-making and scientific understanding. For example, understanding how unmeasured confounding can change estimates about the impact of emissions on cloud properties can help to modify global warming projection models to account for the uncertainty it induces.

We present a novel marginal sensitivity model for continuous treatment effects. This model is used to develop a method that gives the user a corresponding interval representing the "ignorance region" of the possible treatment outcomes per covariate and treatment level [D'A19] for a specified level of violation of the no-hidden confounding assumption. We adapt prior work [Tan06, KMZ19, JMGS21] to the technical challenge presented by continuous treatments. Specifically, we modify the existing model to work with propensity score densities instead of propensity score probabilities (see Section 3 below) and propose a method to relate ignorability violations to the unexplained range of outcomes. Further, we derive bootstrapped uncertainty intervals for the estimated ignorance regions and show how to efficiently compute the intervals, thus providing a method for quantifying the uncertainty presented by finite data and possible violations of the positivity assumption. We validate our methods on synthetic data and provide an application on real-world satellite observations of the effects of anthropogenic emissions on cloud properties. For this application, we develop a new neural network architecture for estimating continuous treatment effects that can take into account spatiotemporal covariates. We find that the model accurately captures known patterns of cloud deepening in response to anthropogenic emission loading with realistic intervals of uncertainty due to unmodeled confounders in the satellite data.

## 2    Problem Setting

Let the random variable $\mathbf{X} \in \mathcal{X}$ model observable covariates. For clarity, we will assume that $\mathcal{X}$ is a $d$-dimensional continuous space: $\mathcal{X} \subseteq \mathbb{R}^d$, but this does not preclude more diverse spaces. Instances of $\mathbf{X}$ are denoted by $\mathbf{x}$. The observable continuous treatment variable is modeled as the random variable $\mathrm{T} \in \mathcal{T} \subseteq \mathbb{R}$. Instances of $\mathrm{T}$ are denoted by $\mathrm{t}$. Let the random variable $\mathrm{Y} \in \mathcal{Y} \subseteq \mathbb{R}$ model the observable continuous outcome variable. Instances of $\mathrm{Y}$ are denoted by $\mathrm{y}$. Using the Neyman-Rubin potential outcomes framework [Ney23, Rub74, Sek08], we model the potential outcome of a treatment level $\mathrm{t}$ by the random variable $\mathrm{Y_t} \in \mathcal{Y}$. Instances of $\mathrm{Y_t}$ are denoted by $\mathrm{y_t}$. We assume that the observational data, $\mathcal{D}_n$, consists of $n$ realizations of the random variables, $\mathcal{D}_n = \{(\mathbf{x}_i, \mathrm{t}_i, \mathrm{y}_i)\}_{i=1}^n$. We let the observed outcome be the potential outcome of the assigned treatment level, $\mathrm{y}_i = \mathrm{y}_{\mathrm{t}_i}$, thus assuming non-interference and consistency [Rub80]. Moreover, we assume that the tuple $(\mathbf{x}_i, \mathrm{t}_i, \mathrm{y}_i)$ are i.i.d. samples from the joint distribution $P(\mathbf{X}, \mathrm{T}, \mathrm{Y_T})$, where $\mathrm{Y_T} = \{\mathrm{Y_t} : \mathrm{t} \in \mathcal{T}\}$.

We are interested in the **conditional average potential outcome (CAPO)** function, $\mu(\mathbf{x}, \mathrm{t})$, and the **average potential outcome (APO)** — or dose-response function — $\mu(\mathrm{t})$, for continuous valued treatments. These functions are defined by the expectations:

$$\mu(\mathbf{x}, \mathrm{t}) \coloneqq \mathbb{E}\left[\mathrm{Y_t} \mid \mathbf{X} = \mathbf{x}\right] \qquad (1) \qquad \qquad \mu(\mathrm{t}) \coloneqq \mathbb{E}\left[\mu(\mathbf{X}, \mathrm{t})\right]. \qquad (2)$$

Under the assumptions of ignorability, $\mathrm{Y_T} \perp\!\!\!\perp \mathrm{T} \mid \mathbf{X}$, and positivity, $p(\mathrm{t} \mid \mathbf{X} = \mathbf{x}) > 0 : \forall \mathrm{t} \in \mathcal{T}, \forall \mathbf{x} \in \mathcal{X}$ — jointly known as *strong ignorability* [RR83] — the CAPO and APO are identifiable from the observational distribution $P(\mathbf{X}, \mathrm{T}, \mathrm{Y_T})$ as:

$$\widetilde{\mu}(\mathbf{x}, \mathrm{t}) = \mathbb{E}\left[\mathrm{Y} \mid \mathrm{T} = \mathrm{t}, \mathbf{X} = \mathbf{x}\right] \qquad (3) \qquad \qquad \widetilde{\mu}(\mathrm{t}) = \mathbb{E}\left[\widetilde{\mu}(\mathbf{X}, \mathrm{t})\right]. \qquad (4)$$

In practice, however, these assumptions rarely hold. For example, there will almost always be unobserved confounding variables, thus violating the ignorability (also known as unconfoundedness or exogeneity) assumption, $\mathrm{Y_T} \not\!\perp\!\!\!\perp \mathrm{T} \mid \mathbf{X}$. Moreover, due to both the finite sample of observed data, $\mathcal{D}$, and also the continuity of treatment $\mathrm{T}$, there will most certainly be values, $\mathrm{T} = \mathrm{t}$, that are unobserved for a given covariate measurement, $\mathbf{X} = \mathbf{x}$, leading to violations or near violations of the positivity assumption (also known as overlap).

## 3 Methods

We propose the continuous marginal sensitivity model (CMSM) as a new marginal sensitivity model (MSM [Tan06]) for continuous treatment variables. The set of conditional distributions of the potential outcomes given the observed treatment assigned, $\{P(Y_t \mid T = t, \mathbf{X} = \mathbf{x}) : t \in \mathcal{T}\}$, are identifiable from data, $\mathcal{D}$. But, the set of marginal distributions of the potential outcomes, $\{P(Y_t \mid, \mathbf{X} = \mathbf{x}) : t \in \mathcal{T}\}$, each given as a continuous mixture,

$$P(Y_t \mid \mathbf{X} = \mathbf{x}) = \int_{\mathcal{T}} p(t' \mid \mathbf{x}) P(Y_t \mid T = t', \mathbf{X} = \mathbf{x}) dt',$$

are not. This is due to the general unidentifiability of the component distributions, $P(Y_t \mid T = t', \mathbf{X} = \mathbf{x})$, where $Y_t$ cannot be observed for units under treatment level $T = t'$ for $t' \neq t$: the well-known "fundamental problem of causal inference" [Hol86]. Yet, under the ignorability assumption, the factual $P(Y_t \mid T = t, \mathbf{X} = \mathbf{x})$ and counterfactual $P(Y_t \mid T = t', \mathbf{X} = \mathbf{x})$ are equal for all $t' \in \mathcal{T}$. Thus, $P(Y_t \mid \mathbf{X} = \mathbf{x})$ and $P(Y_t \mid T = t, \mathbf{X} = \mathbf{x})$ are identical, and any divergence between them is indicative of hidden confounding. But, such divergence is not observable in practice.

The CMSM supposes a degree of divergence between the unidentifiable $P(Y_t \mid \mathbf{X} = \mathbf{x})$ and the identifiable $P(Y_t \mid T = t, \mathbf{X} = \mathbf{x})$ by assuming that the rate of change of $P(Y_t \mid \mathbf{X} = \mathbf{x})$ with respect to $P(Y_t \mid T = t, \mathbf{X} = \mathbf{x})$ is bounded by some value greater than or equal to 1. The Radon-Nikodym derivative formulates the divergence, $\lambda(y_t; \mathbf{x}, t) = \frac{dP(Y_t \mid \mathbf{X}=\mathbf{x})}{dP(Y_t \mid T=t, \mathbf{X}=\mathbf{x})}$, under the assumption that $P(Y_t \mid \mathbf{X} = \mathbf{x})$ is absolutely continuous with respect to $P(Y_t \mid T = t, \mathbf{X} = \mathbf{x}), \forall t \in \mathcal{T}$.

**Proposition 1.** *Under the additional assumption that $P(Y_t \mid T = t, \mathbf{X} = \mathbf{x})$ and the Lebesgue measure are mutually absolutely continuous, the Radon-Nikodym derivative above is equal to the ratio between the unidentifiable "complete" propensity density for treatment $p(t \mid y_t, \mathbf{x})$ and the identifiable "nominal" propensity density for treatment $p(t \mid \mathbf{x})$,*

$$\lambda(y_t; \mathbf{x}, t) = \frac{p(t \mid \mathbf{x})}{p(t \mid y_t, \mathbf{x})}, \tag{5}$$

*Proof (Appendix A.3) and an analysis of this proposition are given in Appendix A.*

The value $\lambda(y_t; \mathbf{x}, t)$ cannot be identified from the observational data alone; the merit of the CMSM is that enables a domain expert to express their belief in what is a plausible degree hidden confounding through the parameter $\Lambda \geq 1$. Where, $\Lambda^{-1} \leq p(t \mid \mathbf{x})/p(t \mid y_t, \mathbf{x}) \leq \Lambda$, reflects a hypothesis that the "complete", unidentifiable propensity density for subjects with covariates $\mathbf{X} = \mathbf{x}$ can be different from the identifiable "nominal" propensity density by at most a factor of $\Lambda$. These inequalities allow for the specification of user hypothesized complete propensity density functions, $p(t \mid y, \mathbf{x})$, and we define the CMSM as the set of such functions that agree with the inequalities.

**Definition 1.** *Continuous Marginal Sensitivity Model (CMSM)*

$$\mathcal{P}(\Lambda) := \left\{ p(t \mid y, \mathbf{x}) : \frac{1}{\Lambda} \leq \frac{p(t \mid \mathbf{x})}{p(t \mid y_t, \mathbf{x})} \leq \Lambda, \forall y \in \mathbb{R}, \forall \mathbf{x} \in \mathcal{X} \right\} \tag{6}$$

**Remark.** Note that the CMSM is defined in terms of a *density ratio*, $p(t \mid \mathbf{x})/p(t \mid y_t, \mathbf{x})$, whereas the MSM for binary-valued treatments is defined in terms of an *odds ratio*, $\frac{P(t \mid \mathbf{x})}{(1-P(t \mid \mathbf{x}))} / \frac{P(t \mid y_t, \mathbf{x})}{(1-P(t \mid y_t, \mathbf{x}))}$. Importantly, naively substituting densities into the MSM for binary-treatments would violate the condition that $\lambda > 0$ as the densities $p(t \mid \mathbf{x})$ or $p(t \mid y_t, \mathbf{x})$ can each be greater than one, which would result in a negative $1 - p(t \mid \cdot)$. The odds ratio is familiar to practitioners. The density ratio is less so. We offer a transformation of the sensitivity analysis parameter $\Lambda$ in terms of the unexplained range of the outcome later.

### 3.1 Continuous Treatment Effect Bounds Without Ignorability

The CAPO and APO (dose-response) functions cannot be point identified from observational data without ignorability. Under the CMSM with a given $\Lambda$, we can only identify a set of CAPO and APO functions jointly consistent with the observational data $\mathcal{D}$ and the continuous marginal sensitivity model. All of the functions in this set are possible from the point of view of the observational data

alone. So to cover the range of all possible functional values, we seek an interval function that maps covariate values, $\mathbf{X} = \mathbf{x}$, to the upper and lower bounds of this set for every treatment value, $t$.

For $t \in \mathcal{T}$ and $\mathbf{x} \in \mathcal{X}$, let $p(y_t \mid t, \mathbf{x})$ denote the density of the distribution $P(Y_t \mid T = t, \mathbf{X} = \mathbf{x})$. As a reminder, this distribution is identifiable from observational data, but without further assumptions the CAPO, $\mu(\mathbf{x}, t) = \mathbb{E}[Y_t \mid \mathbf{X} = \mathbf{x}]$, is not. We can express the CAPO in terms of its identifiable and unidentifiable components as

$$\mu(\mathbf{x}, t) = \frac{\int_{\mathcal{Y}} y_t \frac{p(y_t \mid t, \mathbf{x})}{p(t \mid y_t, \mathbf{x})} dy_t}{\int_{\mathcal{Y}} \frac{p(y_t \mid t, \mathbf{x})}{p(t \mid y_t, \mathbf{x})} dy_t} = \widetilde{\mu}(\mathbf{x}, t) + \frac{\int_{\mathcal{Y}} w(y, \mathbf{x})(y - \widetilde{\mu}(\mathbf{x}, t))p(y \mid t, \mathbf{x})dy}{(\Lambda^2 - 1)^{-1} + \int_{\mathcal{Y}} w(y, \mathbf{x})p(y \mid t, \mathbf{x})dy}, \quad (7)$$
$$\equiv \mu(w(y, \mathbf{x}); \mathbf{x}, t, \Lambda)$$

where, by a one-to-one change of variables, $\frac{1}{p(t \mid y_t, \mathbf{x})} = \frac{1}{\Lambda p(t \mid \mathbf{x})} + w(y, \mathbf{x})(\frac{\Lambda}{p(t \mid \mathbf{x})} - \frac{1}{\Lambda p(t \mid \mathbf{x})})$ with $w : \mathcal{Y} \times \mathcal{X} \to [0, 1]$. Both [KMZ19] and later [JMGS21] provide analogous expressions for the CAPO in the discrete treatment regime under the MSM, and we provide our derivation in Lemma 1.

The uncertainty set that includes all possible values of $w(y, \mathbf{x})$ that agree with the CMSM, *i.e.*, the set of functions that violate ignorability by no more than $\Lambda$, can now be expressed as $\mathcal{W} = \{w : w(y, \mathbf{x}) \in [0, 1] \quad \forall y \in \mathcal{Y}, \forall \mathbf{x} \in \mathcal{X}\}$.

With this set of functions, we can now define the CAPO and APO bounds under the CMSM. The CAPO lower, $\underline{\mu}(\mathbf{x}, t; \Lambda)$, and upper, $\overline{\mu}(\mathbf{x}, t; \Lambda)$, bounds under the CMSM with parameter $\Lambda$ are:

$$\underline{\mu}(\mathbf{x}, t; \Lambda) := \inf_{w \in \mathcal{W}} \mu(w(y, \mathbf{x}); \mathbf{x}, t, \Lambda) \qquad \overline{\mu}(\mathbf{x}, t; \Lambda) := \sup_{w \in \mathcal{W}} \mu(w(y, \mathbf{x}); \mathbf{x}, t, \Lambda)$$
$$= \inf_{w \in \mathcal{W}_{\text{ni}}^H} \mu(w(y); \mathbf{x}, t, \Lambda) \quad (8) \qquad = \sup_{w \in \mathcal{W}_{\text{nd}}^H} \mu(w(y); \mathbf{x}, t, \Lambda) \quad (9)$$

Where the sets $\mathcal{W}_{\text{ni}}^H = \{w : w(y) = H(y_H - y)\}_{y_H \in \mathcal{Y}}$, and $\mathcal{W}_{\text{nd}}^H = \{w : w(y) = H(y - y_H)\}_{y_H \in \mathcal{Y}}$, and $H(\cdot)$ is the Heaviside step function. Lemma 2 in appendix D proves the equivalence in eq. (9) for bounded $Y$. The equivalence in eq. (8) can be proved analogously.

The APO lower, $\underline{\mu}(t; \Lambda)$, and upper, $\overline{\mu}(t; \Lambda)$, bounds under the CMSM with parameter $\Lambda$ are:

$$\underline{\mu}(t; \Lambda) := \mathbb{E}\left[\underline{\mu}(\mathbf{X}, t; \Lambda)\right] \qquad (10) \qquad \overline{\mu}(t; \Lambda) := \mathbb{E}\left[\overline{\mu}(\mathbf{X}, t; \Lambda)\right] \qquad (11)$$

**Remark.** It is worth pausing here and breaking down Equation (7) to get an intuitive sense of how the specification of $\Lambda$ in the CMSM affects the bounds on the causal estimands. When $\Lambda \to 1$, then the $(\Lambda^2 - 1)^{-1}$ term (and thus the denominator) in Equation (7) tends to infinity. As a result, the CAPO under $\Lambda$ converges to the empirical estimate of the CAPO — $\mu(w(y); \mathbf{x}, t, \Lambda \to 1) \to \widetilde{\mu}(\mathbf{x}, t)$ — as expected. Thus, the supremum and infimum in Equations (8) and (9) become independent of $w$, and the ignorance intervals concentrate on point estimates. Next, consider complete relaxation of the ignorability assumption, $\Lambda \to \infty$. Then, the $(\Lambda^2 - 1)^{-1}$ term tends to zero, and we are left with,

$$\mu(w; \cdot, \Lambda \to \infty) \to \widetilde{\mu}(\mathbf{x}, t) + \frac{\int_{\mathcal{Y}} w(y)(y - \widetilde{\mu}(\mathbf{x}, t))p(y \mid t, \mathbf{x})dy}{\int_{\mathcal{Y}} w(y)p(y \mid t, \mathbf{x})dy}, = \widetilde{\mu}(\mathbf{x}, t) + \mathbb{E}_{p(w(y) \mid \mathbf{x}, t)}[Y - \widetilde{\mu}(\mathbf{x}, t)],$$

where, $p(w(y) \mid \mathbf{x}, t) \equiv \frac{w(y)p(y \mid t, \mathbf{x})}{\int_{\mathcal{Y}} w(y')p(y' \mid t, \mathbf{x})dy'}$, a distribution over $Y$ given $\mathbf{X} = x$ and $T = t$. Thus, when we *relax* the ignorability assumption entirely, the CAPO can be anywhere in the range of $Y$.

The parameter $\Lambda$ relates to the proportion of unexplained range in $Y$ assumed to come from unobserved confounders after observing $\mathbf{x}$ and $t$. When a user sets $\Lambda$ to 1, they assume that the entire unexplained range of $Y$ comes from unknown mechanisms independent of $T$. As the user increases $\Lambda$, they attribute some of the unexplained range of $Y$ to mechanisms causally connected to $T$. For bounded $Y_t$, this proportion can be calculated as:

$$\rho(\mathbf{x}, t; \Lambda) := \frac{\overline{\mu}(\mathbf{x}, t; \Lambda) - \underline{\mu}(\mathbf{x}, t; \Lambda)}{\overline{\mu}(\mathbf{x}, t; \Lambda \to \infty) - \underline{\mu}(\mathbf{x}, t; \Lambda \to \infty)} = \frac{\overline{\mu}(\mathbf{x}, t; \Lambda) - \underline{\mu}(\mathbf{x}, t; \Lambda)}{y_{\max} - y_{\min} \mid \mathbf{X} = x, T = t}.$$

The user can sweep over a set of $\Lambda$ values and report the bounds corresponding to a $\rho$ value they deem tolerable (e.g., $\rho = 0.5$ yields bounds for the assumption that half the unexplained range in $Y$ is due

to unobserved confounders). For unbounded outcomes, the limits can be estimated empirically by increasing $\Lambda$ to a large value. Refer to Figure 10 in the appendix for a comparison between $\rho$ and $\Lambda$.

For another way to interpret $\Lambda$, in Appendix A.3.1 we $\Lambda$ can be presented as a bound on the Kullback–Leibler divergence between the nominal and complete propensity scores through the relationship: $|\log(\Lambda)| \geq D_{\mathrm{KL}}(P(Y_t \mid T = t, \mathbf{X} = \mathbf{x})||P(Y_t \mid \mathbf{X} = \mathbf{x}))$.

## 3.2 Semi-Parametric Interval Estimator

Following [JMGS21], we develop a semi-parametric estimator of the bounds in eqs. (8) to (11). Under assumption $\Lambda$, the bounds on the expected potential outcome over $\mu(w(y); \mathbf{x}, t, \Lambda)$ are completely defined in terms of identifiable quantities: namely, the conditional density of the outcome given the assigned treatment and measured covariates, $p(y \mid t, \mathbf{x})$; and the conditional expected outcome $\widetilde{\mu}(\mathbf{x}, t)$. Thus, we define a density estimator, $p(y \mid t, \mathbf{x}, \boldsymbol{\theta})$, and estimator, $\mu(\mathbf{x}, t; \boldsymbol{\theta})$, parameterized by instances $\boldsymbol{\theta}$ of the random variable $\boldsymbol{\Theta}$. The choice of density estimator is ultimately up to the user and will depend on the scale of the problem examined and the distribution of the outcome variable Y. In Section 3.5, we will outline how to define appropriate density estimators for high-dimensional, large-sample, continuous-valued treatment problems. Next, we need an estimator of the integrals in $\mu(w(y); \mathbf{x}, t, \Lambda, \boldsymbol{\theta})$, eq. (7). We use Monte-Carlo (MC) integration to estimate the expectation

---

**Algorithm 1** Grid Search Interval Optimizer

**Require:** $\mathbf{x}$ is an instance of $\mathbf{X}$, t is a treatment level to evaluate, $\Lambda$ is a belief in the amount of hidden confounding, $\boldsymbol{\theta}$ are optimized model parameters, $\widehat{\mathcal{Y}}$ is a set of unique values $\{y \sim p(y \mid t, \mathbf{x}, \boldsymbol{\theta})\}$.

1: **function** GRIDSEARCH($\mathbf{x}, t, \Lambda, \boldsymbol{\theta}, \widehat{\mathcal{Y}}$)
2: $\quad \overline{\mu} \leftarrow -\infty, \overline{y} \leftarrow 0$
3: $\quad \underline{\mu} \leftarrow \infty, \underline{y} \leftarrow 0$
4: $\quad$ **for** $y_H \in \widehat{\mathcal{Y}}$ **do**
5: $\quad\quad \overline{\kappa} \leftarrow \mu(H(y - y_H); \mathbf{x}, t, \Lambda, \boldsymbol{\theta})$
6: $\quad\quad \underline{\kappa} \leftarrow \mu(H(y_H - y); \mathbf{x}, t, \Lambda, \boldsymbol{\theta})$
7: $\quad\quad$ **if** $\overline{\kappa} > \overline{\mu}$ **then**
8: $\quad\quad\quad \overline{\mu} \leftarrow \overline{\kappa}, \overline{y} \leftarrow y_H$
9: $\quad\quad$ **if** $\underline{\kappa} < \underline{\mu}$ **then**
10: $\quad\quad\quad \underline{\mu} \leftarrow \underline{\kappa}, \underline{y} \leftarrow y_H$
11: $\quad$ **return** $\underline{y}, \overline{y}$

---

of arbitrary functions $h(y)$ with respect to the parametric density estimate $p(y \mid t, \mathbf{x}, \boldsymbol{\theta})$: $I(h(y)) := \frac{1}{m} \sum_{i=1}^m h(y_i), \quad y_i \sim p(y \mid t, \mathbf{x}, \boldsymbol{\theta})$. We outline how the Gauss-Hermite quadrature rule is an alternate estimator of these expectations in Appendix C. The integral estimators allow for the semi-parametric estimators for the CAPO and APO bounds under the CMSM to be defined.

The semi-parametric CAPO bound estimators under the CMSM with sensitivity parameter $\Lambda$ are:

$$\underline{\mu}(\mathbf{x}, t; \Lambda, \boldsymbol{\theta}) := \inf_{w \in \mathcal{W}_{\mathrm{ni}}^H} \mu(w(y); \mathbf{x}, t, \Lambda, \boldsymbol{\theta}) \quad (12) \quad \overline{\mu}(\mathbf{x}, t; \Lambda, \boldsymbol{\theta}) := \sup_{w \in \mathcal{W}_{\mathrm{nd}}^H} \mu(w(y); \mathbf{x}, t, \Lambda, \boldsymbol{\theta}) \quad (13)$$

where,

$$\mu(w(y); \mathbf{x}, t, \Lambda, \boldsymbol{\theta}) \equiv \widetilde{\mu}(\mathbf{x}, t; \boldsymbol{\theta}) + \frac{I(w(y)(y - \widetilde{\mu}(\mathbf{x}, t; \boldsymbol{\theta})))}{(\Lambda^2 - 1)^{-1} + I(w(y))}.$$

The semi-parametric APO bound estimators under the CMSM with sensitivity parameter $\Lambda$ are:

$$\underline{\mu}(t; \Lambda, \boldsymbol{\theta}) := \mathbb{E}\left[\underline{\mu}(\mathbf{X}, t; \Lambda, \boldsymbol{\theta})\right] \quad (14) \quad\quad \overline{\mu}(t; \Lambda, \boldsymbol{\theta}) := \mathbb{E}\left[\overline{\mu}(\mathbf{X}, t; \Lambda, \boldsymbol{\theta})\right] \quad (15)$$

**Theorem 1.** *In the limit of data ($n \to \infty$) and MC samples ($m \to \infty$), for observed ($\mathbf{X} = \mathbf{x}, T = t) \in \mathcal{D}_n$, we assume that $p(y \mid t, \mathbf{x}, \boldsymbol{\theta})$ converges in measure to $p(y \mid t, \mathbf{x})$, $\widetilde{\mu}(\mathbf{x}, t; \boldsymbol{\theta})$ is a consistent estimator of $\widetilde{\mu}(\mathbf{x}, t)$, and $p(t \mid y_t, \mathbf{x})$ is bounded away from 0 uniformly for all $y_t \in \mathcal{Y}$. Then, $\underline{\mu}(\mathbf{x}, t; \Lambda, \boldsymbol{\theta}) \xrightarrow{p} \underline{\mu}(\mathbf{x}, t; \Lambda)$ and $\overline{\mu}(\mathbf{x}, t; \Lambda, \boldsymbol{\theta}) \xrightarrow{p} \overline{\mu}(\mathbf{x}, t; \Lambda)$. Proof in Appendix E.*

## 3.3 Solving for w

We are interested in a scalable algorithm to compute the intervals on the CAPO function, eqs. (12) and (13), and the APO (dose-response) function, eqs. (14) and (15). The need for scalability stems not only from dataset size. The intervals also need to be evaluated for arbitrarily many values of the continuous treatment variable, t, and the sensitivity parameter $\Lambda$. The bounds on the CAPO function can be calculated independently for each instance $\mathbf{x}$, and the limits on the APO are an expectation over the CAPO function bounds.

The upper and lower bounds of the CAPO function under treatment, t, and sensitivity parameter, $\Lambda$, can be estimated for any observed covariate value, $\mathbf{x}$, as

$$\underline{\widehat{\mu}}(\mathbf{x}, t; \Lambda, \boldsymbol{\theta}) := \mu(H(\underline{y} - y); \mathbf{x}, t, \Lambda, \boldsymbol{\theta}),$$

$$\widehat{\overline{\mu}}(\mathbf{x}, t; \Lambda, \boldsymbol{\theta}) := \mu(H(y - \overline{y}); \mathbf{x}, t, \Lambda, \boldsymbol{\theta}),$$

where $\underline{y}$ and $\overline{y}$ are found using Algorithm 1. See Algorithm 2 and Appendix F for optional methods.

The upper and lower bounds for the APO (dose-response) function under treatment $T = t$ and sensitivity parameter $\Lambda$ can be estimated over any set of observed covariates $\mathcal{D}_{\mathbf{x}} = \{\mathbf{x}_i\}_{i=1}^n$, as

$$\underline{\widehat{\mu}}(t; \Lambda, \boldsymbol{\theta}) := \frac{1}{n} \sum_{i=1}^n \underline{\widehat{\mu}}(\mathbf{x}_i, t; \Lambda, \boldsymbol{\theta}), \qquad \widehat{\overline{\mu}}(t; \Lambda, \boldsymbol{\theta}) := \frac{1}{n} \sum_{i=1}^n \widehat{\overline{\mu}}(\mathbf{x}_i, t; \Lambda, \boldsymbol{\theta}), \qquad \mathbf{x}_i \in \mathcal{D}_{\mathbf{x}}.$$

### 3.4 Uncertainty about the Continuous Treatment Effect Interval

Following [ZSB19], [DG21], and [CCN+21], we construct $(1 - \alpha)$ statistical confidence intervals for the upper and lower bounds under the CMSM using the percentile bootstrap estimator. [JMSG20] and [JMGS21] have shown that statistical uncertainty is appropriately high for regions with poor overlap. Let $P_{\mathcal{D}}$ be the empirical distribution of the observed data sample, $\mathcal{D} = \{\mathbf{x}_i, t_i, y_i\}_{i=1}^n = \{\mathbf{S}_i\}_{i=1}^n$. Let $\widehat{P}_{\mathcal{D}} = \{\widehat{\mathcal{D}}_k\}_{k=1}^{n_b}$ be the bootstrap distribution over $n_b$ datasets, $\widehat{\mathcal{D}}_k = \{\widehat{\mathbf{S}}_i\}_{i=1}^n$, sampled with replacement from the empirical distribution, $P_{\mathcal{D}}$. Let $Q_\alpha$ be the $\alpha$-quantile of $\mu(w(y); \mathbf{x}, t, \Lambda, \boldsymbol{\theta})$ in the bootstrap resampling distribution: $Q_\alpha := \inf_{\mu^*} \left\{ \widehat{P}_{\mathcal{D}}(\mu(w(y); \mathbf{x}, t, \Lambda, \boldsymbol{\theta}) \leq \mu^*) \geq \alpha \right\}$. Finally, let $\boldsymbol{\theta}_k$ be the parameters of the model of the $k$-th bootstrap sample of the data. Then, the bootstrap confidence interval of the upper and lower bounds of the CAPO function under the CMSM is given by: $\mathrm{CI}_b\left(\mu(\mathbf{x}, t; \Lambda, \alpha)\right) := \left[\underline{\mu}_b(\mathbf{x}, t; \Lambda, \alpha), \overline{\mu}_b(\mathbf{x}, t; \Lambda, \alpha)\right]$, where,

$$\underline{\mu}_b(\mathbf{x}, t; \Lambda, \alpha) = Q_{\alpha/2}\left(\left\{\underline{\widehat{\mu}}(\mathbf{x}, t; \Lambda, \boldsymbol{\theta}_k)\right\}_{k=1}^b\right), \ \overline{\mu}_b(\mathbf{x}, t; \Lambda, \alpha) = Q_{1-\alpha/2}\left(\left\{\widehat{\overline{\mu}}(\mathbf{x}, t; \Lambda, \boldsymbol{\theta}_k)\right\}_{k=1}^b\right).$$

Furthermore, the bootstrap confidence interval of the upper and lower bounds of the APO (dose-response) function under the CMSM are given by: $\mathrm{CI}_b\left(\mu(t; \Lambda, \alpha)\right) := \left[\underline{\mu}_b(t; \Lambda, \alpha), \overline{\mu}_b(t; \Lambda, \alpha)\right]$, where,

$$\underline{\mu}_b(t; \Lambda, \alpha) = Q_{\alpha/2}\left(\left\{\underline{\widehat{\mu}}(t; \Lambda, \boldsymbol{\theta}_k)\right\}_{k=1}^b\right), \qquad \overline{\mu}_b(t; \Lambda, \alpha) = Q_{1-\alpha/2}\left(\left\{\widehat{\overline{\mu}}(t; \Lambda, \boldsymbol{\theta}_k)\right\}_{k=1}^b\right).$$

### 3.5 Scalable Continuous Treatment Effect Estimation

Following [SJS17], [SLB+20], and [NYLN21], we propose using neural-network architectures with two basic components: a feature extractor, $\phi(\mathbf{x}; \boldsymbol{\theta})$ ($\phi$, for short) and a conditional outcome prediction block $f(\phi, t; \boldsymbol{\theta})$. The feature extractor design will be problem and data specific. In Section 5, we look at using both a simple feed-forward neural network, and also a transformer [VSP+17]. For the conditional outcome block, we depart from more complex structures ([SLB+20, NYLN21]) and simply focus on a residual [HZRS16], feed-forward, S-learner [KSBY19] structure. For the final piece of the puzzle, we follow [JMGS21] and propose a $n_y$ component Gaussian mixture density:

$$p(y \mid t, \mathbf{x}, \boldsymbol{\theta}) = \sum_{j=1}^{n_y} \widetilde{\pi}_j(\phi, t; \boldsymbol{\theta}) \mathcal{N}\left(y \mid \widetilde{\mu}_j(\phi, t; \boldsymbol{\theta}), \widetilde{\sigma}_j^2(\phi, t; \boldsymbol{\theta})\right),$$

and $\widetilde{\mu}(\mathbf{x}, t; \boldsymbol{\theta}) = \sum_{j=1}^{n_y} \widetilde{\pi}_j(\phi, t; \boldsymbol{\theta}) \widetilde{\mu}_j(\phi, t; \boldsymbol{\theta})$ [Bis94]. Models are optimized by maximizing the log-likelihood of $p(y \mid t, \mathbf{x}, \boldsymbol{\theta})$.

## 4 Related Works

**Scalable Continuous Treatment Effect Estimation.** Using neural networks to provide scalable solutions for estimating the effects of continuous-valued interventions has received significant

attention in recent years. [BJvdS20] provide a Generative Adversarial Network (GAN) approach. The dose-response network (DRNet) [SLB$^+$20] provides a more direct adaptation of the TarNet [SJS17] architecture for continuous treatments. The varying coefficient network VCNet [NYLN21] generalizes the DRNet approach and provides a formal result for incorporating the target regularization technique presented by [SBV19]. The RieszNet [CCN$^+$21] provides an alternative approach for targeted regularization. Adaptation of each method is straightforward for use in our sensitivity analysis framework by replacing the outcome prediction head of the model with a suitable density estimator.

**Sensitivity and Uncertainty Analyses for Continuous Treatment Effects.** The prior literature for continuous-valued treatments has focused largely on parametric methods assuming linear treatment/outcome, hidden-confounder/treatment, and hidden-confounder/outcome relationships [CHH16, DHCH16, MSDH16, Ost19, CH20a, CH20b]. In addition to linearity, these parametric methods need to assume the structure and distribution of the unobserved confounding variable(s). [CKC$^+$19] allows for sensitivity analysis for arbitrary structural causal models under the linearity assumption. The MSM relaxes both the distributional and linearity assumptions, as does our CMSM extension. A two-parameter sensitivity model based on Riesz-Frechet representations of the target functionals, here the APO and CAPO, is proposed by [CCN$^+$21] as a way to incorporate confidence intervals and sensitivity bounds. In contrast, we use the theoretical background of the marginal sensitivity model to derive a one-parameter sensitivity model. [DBSC21] purport to quantify the bias induced by unobserved confounding in the effects of continuous-valued interventions, but they do not present a formal sensitivity analysis. Simultaneously and independently of this work, [MVSG] are deriving a sensitivity model that bounds the partial derivative of the log density ratio between complete and nominal propensity densities. Bounding the effects of continuous valued interventions has also been explored using instrumental variable models [KKS20, HWZW21, PZW$^+$22].

## 5 Experiments

Here we empirically validate our method. First, we consider a synthetic structural causal model (SCM) to demonstrate the validity of our method. Next, we show the scalability of our methods by applying them to a real-world climate-science-inspired problem. Implementation details (appendix H), datasets (appendix G), and code are provided at `https://github.com/oatml/overcast`.

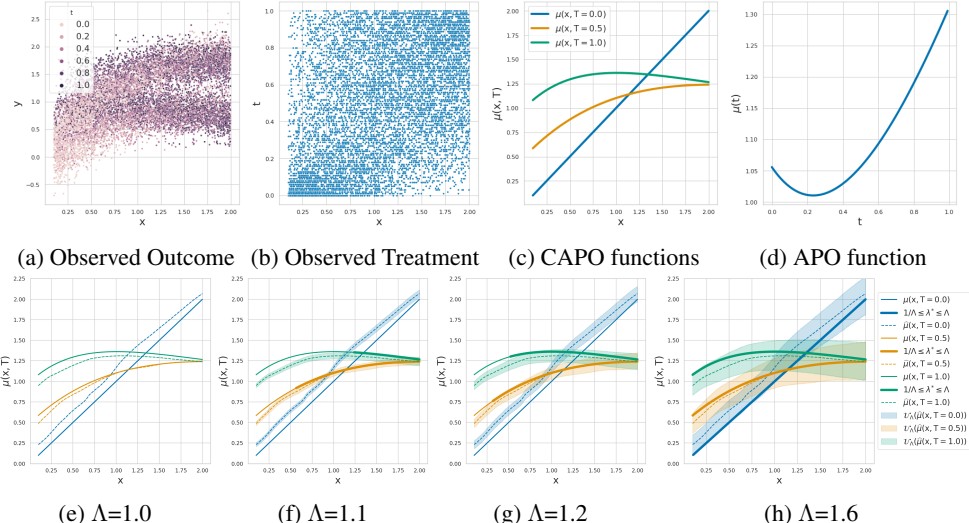

(a) Observed Outcome    (b) Observed Treatment    (c) CAPO functions    (d) APO function

(e) Λ=1.0      (f) Λ=1.1      (g) Λ=1.2      (h) Λ=1.6

Figure 1: Figures 1a to 1d: Synthetic data and ground truth functions. Figures 1e to 1h Causal uncertainty under hypothesized Λ values. Solid lines are ground truth; thick solid lines where the true $\lambda^*$ is within the range of hypothesized Λ, thin solid lines otherwise. The dotted lines are the estimated CAPO. Shaded regions are estimated CMSM intervals.

### 5.1 Synthetic

Figure 1 presents the synthetic dataset (additional details about the SCM are given in Appendix G.1). Figure 1a plots the observed outcomes, y, against the observed confounding covariate, x. Each

datapoint is colored by the magnitude of the observed treatment, t. The binary unobserved confounder, u, induces a bi-modal distribution in the outcome variable, y, at each measured value, x. Figure 1b plots the assigned treatment, t, against the observed confounding covariate, x. We can see that the coverage of observed treatments, t, varies for each value of x. For example, there is uniform coverage at $X = 1$, but low coverage for high treatment values at $X = 0.1$, and low coverage for low treatment values at $X = 2.0$. Figure 1c plots the true CAPO function over the domain of observed confounding variable, X, for several values of treatment ($T = 0.0$, $T = 0.5$, and $T = 1.0$). For lower magnitude treatments, t, the CAPO function becomes more linear, and for higher values, we see more effect heterogeneity and attenuation of the effect size as seen from the slope of the CAPO curve for $T = 0.5$ and $T = 1.0$. Figure 1d plots the the APO function over the domain of the treatment variable T.

**Causal Uncertainty** We want to show that in the limit of large samples (we set $n$ to $100k$), the bounds on the CAPO and APO functions under the CMSM include the ground truth when the CMSM is correctly specified. That is, when $1/\Lambda \leq \lambda^*(t, x, u) \leq \Lambda$, for user specified parameter $\Lambda$, the estimated intervals should cover the true CAPO or APO. This is somewhat challenging to demonstrate as the true density ratio $\lambda^*(t, x, u)$ (eq. (50)), varies with t, x, and u. Figures 1e to 1h work towards communicating this. In Figure 1e, we see that each predicted CAPO function (dashed lines) is biased away from the true CAPO functions (solid lines). We use thick solid lines to indicate cases where $1/\Lambda \leq \lambda^*(t, x, u) \leq \Lambda$, and thin solid lines otherwise. Therefore thick solid lines indicate areas where we expect the causal intervals to cover the true functions. Under the erroneous assumption of ignorability ($\Lambda = 1$), the CMSM bounds have no width. In Figure 1f, we see that as we relax our ignorability assumption ($\Lambda = 1.1$) the intervals (shaded regions) start to grow. Note the thicker orange line: this indicates that for observed data described by $X > 0.5$ and $T = 0.5$, the actual density ratio is in the bounds of the CMSM with parameter $\Lambda = 0.5$. We see that our predicted bounds cover the actual CAPO function for these values. We see our bounds grow again in Figure 1g when we increase $\Lambda$ to 1.2. We see that more data points have $\lambda^*$ values that lie in the CMSM range and that our bounds cover the actual CAPO function for these values. In Figure 1h we again increase the parameter of the CMSM. We see that the bounds grow again, and cover the true CAPO functions for all of the data that satisfy $1/\Lambda \leq \lambda^*(t, x, u) \leq \Lambda$.

**Statistical Uncertainty** Now we relax the infinite data assumption and set $n = 1000$. This decrease in data will increase the estimator error for the CAPO and APO functions. So the estimated functions will not only be biased due to hidden confounding, but they may also be erroneous due to finite sample variance. We show this in Figure 2b where the blue dashed line deviates from the actual blue solid line as **x** increases beyond 1.0. However, Figure 2b shows that under the correct CMSM, the uncertainty aware confidence intervals,

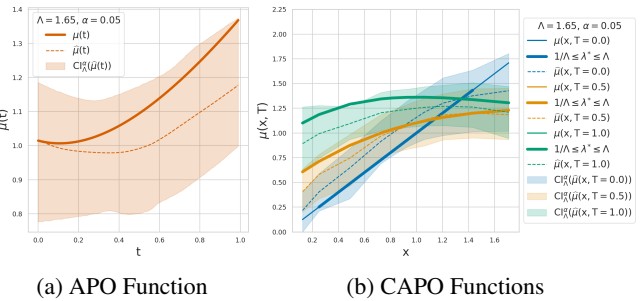

(a) APO Function  (b) CAPO Functions

Figure 2: Statistical and causal uncertainty, $\alpha$ is statistical significance level for the bootstrap. see Figure 1 for other details.

section 3.4, cover the actual CAPO functions for the range of treatments considered. Figure 2a demonstrates that this holds for the APO function as well.

## 5.2 Estimating Aerosol-Cloud-Climate Effects from Satellite Data

**Background** The development of the model above, and the inclusion of treatment as a continuous variable with multiple, unknown confounders, is motivated by a real-life use case for a prime topic in climate science. Aerosol-cloud interactions (ACI) occur when anthropogenic emissions in the form of aerosol enter a cloud and act as cloud condensation nuclei (CCN). An increase in the number of CCN results in a shift in the cloud droplets to smaller sizes which increases the brightness of a cloud and delays precipitation, increasing the cloud's lifetime, extent, and possibly thickness [Two77, Alb89, TCGB17]. However, the magnitude and sign of these effects are heavily dependent on the environmental conditions surrounding the cloud [DL20]. Clouds remain the largest source of uncertainty in our future climate projections [MDZP$^+$21]; it is pivotal to understand how human emissions may be altering their ability to cool. Our current climate models fail to accurately emulate

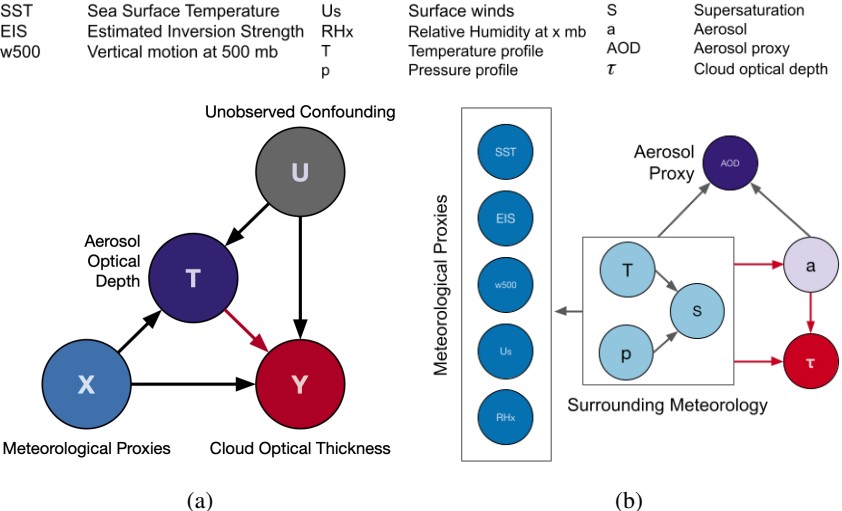

(a)                       (b)

Figure 3: Causal diagrams. Figure 3a, a simplified causal diagram representing what we are reporting within; aerosol optical depth (AOD, regarded as the treatment T) modulates cloud optical depth ($\tau$, Y), which itself is affected by hidden confounders (U) and the meteorological proxies (X). Figure 3b, an expanded causal diagram of ACI. The aerosol (a) and aerosol proxy (AOD), the true confounders (light blue), their proxies (dark blue), and the cloud optical depth (red).

ACI, leading to uncertainty bounds that could offset global warming completely or double the effects of rising $CO_2$ [BRA$^+$13].

**Defining the Causal Relationships** Clouds are integral to multiple components of the climate system, as they produce precipitation, reflect incoming sunlight, and can trap outgoing heat [SF09]. Unfortunately, their interconnectedness often leads to hidden sources of confounding when trying to address how anthropogenic emissions alter cloud properties.

Ideally, we would like to understand the effect of aerosols ($a$) on the cloud optical thickness, denoted $\tau$. However, this is currently impossible. Aerosols come in varying concentrations, chemical compositions, and sizes [SGW$^+$16] and we cannot measure these variables directly. Therefore, we use aerosol optical depth (AOD) as a continuous, 1-dimensional proxy for aerosols. Figure 3b accounts for the known fact that AOD is an imperfect proxy impacted by its surrounding meteorological environment [CNP$^+$17]. The meteorological environment is also a confounder that impacts cloud thickness $\tau$ and aerosol concentration $a$. Additionally, we depend on simulations of the current environment in the form of reanalysis to serve as its proxy.

Here we report AOD as a continuous treatment and the environmental variables as covariates. However, aerosol is the actual treatment, and AOD is only a confounded, imperfect proxy (Figure 3a). This model cannot accurately capture all causal effects and uncertainty due to known and unknown confounding variables. We use this simplified model as a test-bed for the methods developed within this paper and as a demonstration that they can scale to the underlying problem. Future work will tackle the more challenging and realistic causal model shown in Figure 3b, noting that the treatment of interest $a$ is multi-dimensional and cannot be measured directly.

**Model** We use daily observed $1° \times 1°$ means of clouds, aerosol, and the environment from sources shown in Table 1 of Appendix G. To model the spatial correlations between the covariates on a given day, we use multi-headed attention [VSP$^+$17] to define a transformer-based feature extractor. Modeling the spatial dependencies between meteorological variables is motivated by confounding that may be latent in the relationships between neighboring variables. These dependencies are unobserved from the perspective of a single location. This architectural change respects both the assumed causal graph (fig. 3a) and some of the underlying physical causal structure. We see in Figure 4 (Left) that modeling *context* with the transformer architecture significantly increases the predictive accuracy of the model when compared to a simple feed-forward neural network (*no context*). **Discussion & Results** The results for the APO of cloud optical depth ($\tau$) as the "treatment", AOD, increases are shown in Figure 4. As the assumed strength of confounding increases ($\Lambda > 1$), the range of uncertainty

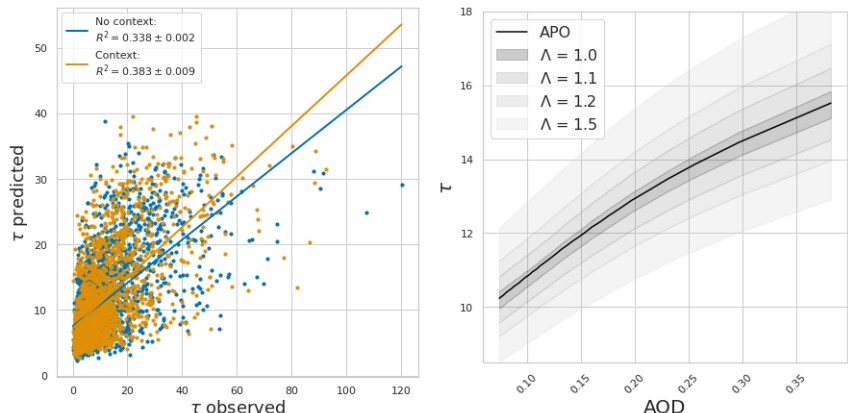

Figure 4: Left: The values of the observed, true $\tau$ against the modeled $\tau$. Right: The curve for continuous treatment outcome of our aerosol proxy (AOD) on cloud optical depth ($\tau$). The darkest shaded region ($\Lambda = 1$) represents the uncertainty in the treatment outcome from the ensemble due to finite data. As the strength of confounders increases ($\Lambda > 1.0$), the range of uncertainty in the treatment outcome increases.

in the treatment outcome increases. Within this range of confounding, the modeled outcomes agree with two conflicting hypotheses. First, that aerosol acts to invigorate the cloud, inducing a large response that would follow a maximum curve within this uncertainty range [CS11, DL21]. And second, that aerosol has little impact on cloud depth, and the actual response is a minimal, flat line [GGS$^+$19]. We further find the reported dose-response curves in agreement with multiple estimates of aerosol-cloud interactions using satellite observations [BTG02, MSJ$^+$07, TCQB19]. The upper bound for $\log \Lambda = .2$ agrees with measurements of the in-cloud environment and aerosol-cloud interactions from aircraft-mounted sensors [PZ13], this may indicate the need for additional control variables when using satellite data.

The resolution of the satellite observations ($1° \times 1°$ daily means) could be averaging various cloud types and obscuring the signal. Future work will investigate how higher resolution (20km $\times$ 20km) data with constraints on cloud type may resolve some confounding influences. However, even our more detailed causal model (Figure 3b) cannot account for all confounders; we expected, and have seen, imperfections in our model of this complex effect. The model's results require further expert validation to interpret the outcomes and uncertainty.

**Societal Impact** Geoengineering of clouds by aerosol seeding could offset some amount of warming due to climate change, but also have disastrous global impacts on weather patterns [DGL$^+$22]. Given the uncertainties involved in understanding aerosol-cloud interactions, it is paramount that policy makers are presented with projected outcomes if a proposals assumptions are wrong or relaxed.

## Acknowledgments and Disclosure of Funding

We would like to thank Angela Zhou for introducing us to the works of [ZSB19] and [DG21]. These works use the percentile bootstrap for finite sample uncertainty estimation within their sensitivity analysis methods. We would also like to thank Lewis Smith for helping us understand the Marginal Sensitivity Model of [Tan06] in detail. Finally, we would like to thank Clare Lyle and all anonymous reviewers for their valuable feedback.

This research was supported by the European Research Council (ERC) project constRaining the EffeCts of Aerosols on Precipitation (RECAP) under the European Union's Horizon 2020 research and innovation program with grant agreement no. 724602 and from the European Union's Horizon 2020 research and innovation program project Constrained aerosol forcing for improved climate projections (FORCeS) under grant agreement No 821205. and Marie Skłodowska-Curie grant agreement No 860100 (iMIRACLI). This work used JASMIN, the UK's collaborative data analysis environment (http://jasmin.ac.uk). U.S. was partially supported by the Israel Science Foundation (grant No. 1950/19).

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
