# OpenReview forum: "Scalable Sensitivity and Uncertainty Analyses for Causal-Effect Estimates of Continuous-Valued Interventions"
_NeurIPS.cc/2022/Conference — NeurIPS 2022 Accept_

### Official Review · Reviewer_zMDP · 2022-06-28

**Rating:** 7
**Confidence:** 4
**Soundness:** 4 excellent
**Presentation:** 3 good
**Contribution:** 3 good

**Summary:**

The paper derives bounds for causal treatment effect estimates when the intervention is continuous and there exists a user-specified amount of unobserved confounding. The main innovation is to extend previous bounds which were derived for discrete treatments to the case of continuous treatments. Non-parametric models are assumed for the treatment conditional on the covariates, and for the outcome conditional on the treatment and covariates.

**Questions:**

Given the multiple somewhat overlapping works currently under review, I would suggest expanding the "Sensitivity and Uncertainty Analyses for Continuous Treatment Effects" section in "Related Works" to include more details about the differences of the methods.

Also, I'm wondering if there aren't additional papers published in different fields (other than machine learning), where similar results could have been discussed earlier. On a quick googling, I found at least the following, which, based on title and abstract, might have a similar goal: "Bias formulas for sensitivity analysis of unmeasured confounding for general outcomes, treatments, and confounders" by VanderWeele and Arah, Epidemiology, 2011. Could you clarify the difference of the present work to this, and also check if you can find other similar papers published in, e.g., economics, epidemiology, or (bio)statistics?

Proposition 1 has an assumption that P(Y_t|T=t,X=x) is equivalent to a Lebesgue measure. I read this such that the distribution is uniform, which seems very restrictive. Could you clarify this, please?

I don't follow eq (7). It's probably similar to KMZ19, but it would be good to make the present article self-standing. For example, what is the w(.) function and what is its role here intuitively? It's defined after eq (9) for one argument, but it seems to be used in the text interchangable with one or two arguments.

Similarly to the previous comment, I'm not sure what the y with the bar below/above in Section 3.3 is.


Very minor:
l. 210: D has subscript j while indexing uses k.

l. 230: By mazimizing the log-likelihood, not "minimizing".


**Limitations:**

Yes.

**Strengths And Weaknesses:**

Strengths:

The article has an important and timely topic. The derivations for the bounds seem rigorous, and are validated empirically in the experiments. The analytical results are derived for the asymptotic case, for finite case the article proposes bootstrapping to get a bounds for the bounds. Related works are covered well (though see questions below). The presentation is clear. The real-world application in climate change is important and well-described.

Weaknesses:

This appears a crowded topic, as there seem to be at least two other recently arxiv'ed papers on the same topic (Chernozhukov et al., 2021; Marmarelis et al., 2022), which may differ in some details (I haven't read those papers closely). The idea for the topic seems to have been discussed in previous year's "Causal Inference & Machine Learning: Why now?" Neurips Workshop, which potentially has spawned the interest (my speculation). Anyway, this seems to slightly decrease the novelty of the present article. I am not sure if any of these related papers is more deserving to be published first, but at least the present article I found a rather complete and good package and tentatively can support its acceptance, unless other reviewers identify major issues. I don't find any other major problems.

There were some small problems with presentation, especially not all notation used in the formulas is defined/explained clearly (see below). The figures in the Experiments section are not sharp when printed out. Otherwise the presentation is clear.

---

> ### Author Response · Authors · 2022-08-02
> **Thank you for your review 2/2**
>
> ### "I don't follow eq (7). It's probably similar to KMZ19, but it would be good to make the present article self-standing. For example, what is the w(.) function and what is its role here intuitively? It's defined after eq (9) for one argument, but it seems to be used in the text interchangable with one or two arguments."
>
> It is best to start at the bound in equation (6) to understand the role of $w(\cdot)$:
>
> $\frac{1}{\Lambda p(\mathrm{t} \mid \mathbf{x})} \leq \frac{1}{p(\mathrm{t} \mid \mathrm{y}_{\mathrm{t}}, \mathbf{x})} \leq \frac{\Lambda}{ p(\mathrm{t} \mid \mathbf{x})}$.
>
> The role of $w(\cdot)$ is to express the hypothesized inverse complete propensity density, $\frac{1}{p(\mathrm{t} \mid \mathrm{y}, \mathbf{x})}$, as a linear interpolation between the lower bound, $\frac{1}{\Lambda p(\mathrm{t} \mid \mathbf{x})}$, and the upper bound, $\frac{\Lambda}{ p(\mathrm{t} \mid \mathbf{x})}$. We first define $w(\cdot)$ after equation (7) in general as a function of $\mathbf{x}$ and $\mathrm{y}$ with range $[0, 1]$, such that:
>
> $\frac{1}{p(\mathrm{t} \mid \mathrm{y}, \mathbf{x})} = \frac{1}{\Lambda p(\mathrm{t} \mid \mathbf{x})} + w(\mathrm{y}, \mathbf{x}) \left( \frac{\Lambda}{p(\mathrm{t} \mid \mathbf{x})}-\frac{1}{\Lambda p(\mathrm{t} \mid \mathbf{x})}\right)$.
>
> To get a sense of this interpolation, it is easy to see that when $w(\mathrm{y}, \mathbf{x}) = 0$, the second r.h.s. term disappears leaving just the lower bound, $\frac{1}{\Lambda p(\mathrm{t} \mid \mathbf{x})}$, and when $w(\mathrm{y}, \mathbf{x}) = 1$, the lower bounds cancel out leaving just the upper bound, $\frac{\Lambda}{p(\mathrm{t} \mid \mathbf{x})}$. When $w(\mathrm{y}, \mathbf{x}) \in (0, 1)$, the function can take on any value between the two extrema.
>
> We define $w(\cdot)$ in the most general case because we started looking at a gradient descent approach to solve the bounds on the CAPO and APO functions, which we detail in Appendix F. The gradient descent approach optimizes a parameterized version of $w(\mathrm{y}, \mathbf{x})$. The gradient descent approach may be an interesting avenue for future work, but it did not yield any clear advantage over the grid search approach in our initial analyses. While we relegated the gradient descent approach to the appendix, it may be best to keep the definition general here to inspire other approaches.
>
> In equations (8) and (9), we drop the $\mathbf{x}$ and express $w(\cdot)$ as just a function of $\mathrm{y}$ for two reasons. First, we pick a specific form for $w(\cdot)$ — the Heaviside step function — for conciseness in the lead-up to Theorem 1. Theorem 1 uses Lemma 2, showing that the solution space for $w$ is limited to step functions. Second, given this choice and the fact that the infima and suprema in (8) and (9) are determined independently for each $\mathbf{x}$, the $\mathbf{x}$ notation becomes redundant. We are happy to add a note in the main text and perhaps add this discussion to the appendix.
>
>
> ### "Similarly to the previous comment, I'm not sure what the y with the bar below/above in Section 3.3 is."
>
> The parameters returned by the grid search algorithm (Algorithm 1) for the step function, $H$, are given by $\underline{\mathrm{y}}$ and $\overline{\mathrm{y}}$, where $\underline{\mathrm{y}}$ is the parameter associated with the lower bound estimate, and $\overline{\mathrm{y}}$ is the parameter for the upper bound estimate.
>
> ### "The figures in the Experiments section are not sharp when printed out. Otherwise the presentation is clear." "Very minor: l. 210: D has subscript j while indexing uses k." "l. 230: By mazimizing the log-likelihood, not "minimizing"."
>
> Thank you for pointing these out to us. We will use vector graphics in the camera-ready should this paper be accepted for publication. We have updated the manuscript to address the errors.

---

> > ### Comment · Reviewer_zMDP · 2022-08-08
> > **Thank you for the clarifications.**
> >
> > Thank you for the clarifications. I have no further comments.

---

> > > ### Author Response · Authors · 2022-08-09
> > > **Thank you**
> > >
> > > Thank you again for your feedback and corrections.

---

> ### Author Response · Authors · 2022-08-02
> **Thank you for your review 1/2**
>
> Thank you for your detailed questions and comments. It is encouraging that you find the topic  important and timely. We address your individual comments below.
>
> TL;DR — *We discuss recent related works. We propose to add a detailed literature review to the appendix to include discrete treatment methods. We point out the efforts made to go beyond the machine learning literature. We address your technical questions. We fix the errors you have found.*
>
> ### "This appears a crowded topic, as there seem to be at least two other recently arxiv'ed papers on the same topic (Chernozhukov et al., 2021; Marmarelis et al., 2022)"
>
> We would agree that this is a timely topic and do not believe that the publication of any of these papers should preclude publication of another. Our motivation for developing this method was that we were initially using discrete treatment methods to analyze the climate data. The overwealming initial feedback from the climate science community was that to treat AOD as a continuous variable. While there were methodologies to quantify statistical and causal uncertainty for discrete treatments, there appeared to be a gap in the continuous treatment regime. We became aware of (Chernozhukov et al., 2021) shortly after begining to develop our methodology. Because our methods take different approaches to the sensitivity analysis, we pushed forward on developing our own. We became aware of (Marmarelis et al., 2022) after submission of this work to an earlier conference. They take the MSM approach, but derive a different MSM. It will make for interesting future work to compare and contrast these methods. However, since each are unique and independently developed approaches to a timely problem, we do not see their co-occurence alone as a reason for any to be rejected.
>
> ### "Also, I'm wondering if there aren't additional papers published in different fields (other than machine learning), where similar results could have been discussed earlier. On a quick googling, I found at least the following, which, based on title and abstract, might have a similar goal: "Bias formulas for sensitivity analysis of unmeasured confounding for general outcomes, treatments, and confounders" by VanderWeele and Arah, Epidemiology, 2011. Could you clarify the difference of the present work to this, and also check if you can find other similar papers published in, e.g., economics, epidemiology, or (bio)statistics?"
>
> Thank you for sharing this work. It is a parametric approach in the same vein as CHH16, DHCH16, MSDH16, Ost19, CH20a, and CH20b. However, it looks at discrete rather than continuous treatment levels. We tried earnestly to do an extensive survey outside the machine learning literature. CHH16 is from the Journal of Research on Educational Effectiveness, DHCH16 is from Statistics in Medicine, MSDH16 is from Political Analysis, Ost19 is from the Journal of Business & Economic Statistics, and CH20a is from the Journal of the Royal Statistical Society. We chose to prioritize methods that work with continuous treatment values in the related works due to space constraints. Given your comments — and the comments by Reviewer UYQB and Reviewer eRS3 — it is clear that a more detailed treatment of the related works would be welcome. We would be happy to include a more thorough report in the appendix.
>
> ### "Proposition 1 has an assumption that P(Y_t|T=t,X=x) is equivalent to a Lebesgue measure. I read this such that the distribution is uniform, which seems very restrictive. Could you clarify this, please?"
>
> Equivalence here is in the measure-theoretic sense. Namely, we assume that $P(\mathrm{Y}_\mathrm{t}|\mathrm{T}=\mathrm{t},\mathrm{X}=\mathrm{x})$ is absolutely continuous with respect to the Lebesgue measure, *and* that the Lebesgue measure is absolutely continuous with respect to $P(\mathrm{Y}_\mathrm{t}|\mathrm{T}=\mathrm{t},\mathrm{X}=\mathrm{x})$. This does not imply that $\mathrm{Y}_\mathrm{t}|\mathrm{T}=\mathrm{t},\mathrm{X}=\mathrm{x}$ is uniformly distributed. Rather, it says that the zero measure sets of the Lebesgue measure and $P(\mathrm{Y}_\mathrm{t}|\mathrm{T}=\mathrm{t},\mathrm{X}=\mathrm{x})$ coincide. This assumption certainly holds when $\mathrm{Y}_\mathrm{t}|\mathrm{T}=\mathrm{t},\mathrm{X}=\mathrm{x}$ is distributed with support over the whole real line, for example, with Cauchy, Gaussian, or Laplace distributed random variables. The assumption is also consistent with our use of mixture density networks to model $p(\mathrm{y} | \mathbf{x}, \mathrm{t})$. Admittedly, equivalence is an overloaded term. We have updated the manuscript to make this more explicit (lines 93-94).

---

### Official Review · Reviewer_r7RZ · 2022-07-13

**Rating:** 6
**Confidence:** 3
**Soundness:** 3 good
**Presentation:** 2 fair
**Contribution:** 3 good

**Summary:**

Ignorability assumption is essential to ATE, however, this is often not satisfied in the real world. This means that there will be ignorance in the causal estimates when this assumption is relaxed. The paper addresses this issue by proposing a neural network based model that are scalability for ananlysing sensitivity and uncertainty when the ignorability assumption does not hold true for continuous data.

**Questions:**

- It's not clear to me why the exact formulation of the SCM of the synthetic data was chosen. Would be nice if the authors could elaborate on this.

- Section 3 is a bit difficult to follow, would be nice to have more intutive explanations of the derivations.

**Limitations:**

The authors were transpararent about the limitations of their work, such as lambda may not be easily identified.

**Strengths And Weaknesses:**

Pros,

- The paper addresses an important issue of sensitivity analysis and uncertainty to measure the ignorance in causal estimate when the ignorability assumption is voileted. The paper also proposes a scalable neural network based approach, which can be applicable to a wilder set of data and problems.

- The beignning of the paper is very well written and easy to follow. The settings, assumptions and derivations are clear and concise.

Cons,

- The generation of the synthetic data seems to be in a particular form, it's not clear to me why this exact formulation was chosen. Could the authors explain a little more on this?

- The latter sections (section 3) are slightly difficult to follow. It would be nice to have more intuitive explanation of the derivations throughout the section.

- For the experiment, it's not exactly clear to me how the assumption of ignorability fits into the neural network architecture.

---

> ### Author Response · Authors · 2022-08-02
> **Thank you for your review**
>
> Thank you very much for your feedback. We are glad you found the beginning of the paper clear and intuitive. We recognize that we can improve the overall readability for you. We address your questions below.
>
> TL;DR — *We provide context for the synthetic experiment. We request clarity on how we can improve the readability of Section 3. We explain how ignorability fits in with the neural-network architecture.*
>
> ### "The generation of the synthetic data seems to be in a particular form, it's not clear to me why this exact formulation was chosen. Could the authors explain a little more on this?"
>
> The synthetic data design illustrates several different aspects. We need a ground truth density for the complete propensity to calculate reference $\Lambda$ values for each $\mathrm{x}$ value. These are unique for each $\mathrm{x}$ show we show the bounds growing for increasing lambda to show how we cover the ground truth CAPO at the ground truth $\Lambda$ value. The choice of ground truth complete propensity density induces regions of high and low overlap, and we see the statistical uncertainty grow in those regions. We have the CAPO functions follow a varying non-linear form to make it more challenging for the estimator and reflect real-world possibilities.
>
> ### "The latter sections (section 3) are slightly difficult to follow. It would be nice to have more intuitive explanation of the derivations throughout the section."
>
> Thank you for this comment. Were there specific areas that you had in mind? Were the remarks helpful? We would be happy to include more in sections 3.2-3.5 if we had more space.
>
> ### "For the experiment, it's not exactly clear to me how the assumption of ignorability fits into the neural network architecture."
>
> The ignorability assumption does not factor into the neural network architecture per se. A neural network can be used to estimate the statistical value $\mathbb{E}[\mathrm{Y} \mid \mathbf{x}, \mathrm{t}]$. Under ignorability (and other assumptions), the statistical estimand above is equivalent to the causal effect, $\mathbb{E}[\mathrm{Y}_\mathrm{t} \mid \mathbf{x}]$. Sensitivity analysis allows us to quantify the interval of $\mathbb{E}[\mathrm{Y} \mid \mathbf{x}, \mathrm{t}]$ values that are compatible with the data and a user-specified relaxation of the hidden confounding assumption. Our sensitivity analysis depends not only on the mean value estimate $\mathbb{E}[\mathrm{Y} \mid \mathbf{x}, \mathrm{t}]$, but also the the density of $\mathrm{Y}$ given $\mathbf{X}=\mathbf{x}$ and $\mathrm{T}=\mathrm{t}$, $p(\mathrm{y} | \mathbf{x}, \mathrm{t})$. Where a standard regression neural-network just outputs the mean value estimate, we use a mixture density network to model $p(\mathrm{y} | \mathbf{x}, \mathrm{t})$. Modeling the density allows us to sample $\mathrm{Y}$ values and perform our sensitivity analysis. $\Lambda$ fits in at this stage, after model training at inference time.

---

> ### Author Response · Authors · 2022-08-09
> **Reviewer-author discussion**
>
> Thank you again for taking the time to review our paper. We hope our response below has addressed your concerns. If you have any further suggestions, we would be happy to discuss them with you prior to your review confirmation.

---

### Official Review · Reviewer_UYQB · 2022-07-15

**Rating:** 8
**Confidence:** 4
**Soundness:** 4 excellent
**Presentation:** 4 excellent
**Contribution:** 4 excellent

**Summary:**

In this paper, the authors develop a sensitivity analysis method for estimating causal effects with continuous-valued interventions. This is important because when performing causal inference from observational data, common assumptions like no unobserved confounding rarely hold. Building on a sensitivity analysis formulation for binary treatments, the authors develop a continuous marginal sensitivity model (CMSM) that accounts for possible unobserved confounding by bounding the density ratio between the observational and "complete" treatment propensity distribution. This sensitivity model allows the authors to compute upper and lower bounds on the conditional average potential outcome (which when marginalized over covariates yields the average potential outcome). Thus, for a given value of the sensitivity parameter, and for any value of the treatment and covariates, the method produces an interval for the conditional average potential outcome that can be subsequently used to estimate the range of causal effects compatible with the data. In synthetic experiments the authors validate various aspects of their approach. Finally, they apply their method to a simplified causal question in climate science, showing that a range of hypotheses are compatible with the data under varying degrees of possible unobserved confounding.

**Questions:**

* How does the proposed method address possible positivity violations? The authors allude to this in the introduction but (it seems to me) it is never explicitly spelled out. It seems that the uncertainty quantification via bootstrapping is meant to capture uncertainty in, e.g., regions with poor overlap? I think some explicit statement about this (say, in Section 3.4) would help tie up any loose ends regarding the authors' claims.

* In the climate experiment, what guidance is there for how to pick relevant $\Lambda$ values to consider? Could the authors add some kind of qualitative discussion/judgement here?

**Limitations:**

Yes, the authors adequately address limitations.

**Strengths And Weaknesses:**

I really enjoyed this paper. I think the problem is important, the paper is well-written and well-executed, and the developed methodology seems both elegant and practical. As the authors note, I think sensitivity analysis is very important for bringing the value of causal inference from observational data to policy making in practical scenarios, and I think the current paper represents a useful method for doing so.

I mainly have a few clarifying questions & minor suggestions:
* The main possible difficulty with the proposed method that came to mind is how to select values of the sensitivity parameter. As opposed to the odds ratio (which, as the authors note, is generally interpretable to practitioners), the density ratio is somewhat difficult to make judgements about directly. The authors provide an alternative characterization in terms of the "proportion of unexplained range in $Y$", but even this is (to my knowledge) not a commonly considered statistic. Can this be related in any way to, e.g., the $R^2$ (i.,e., the fraction of *variance* unexplained)? Also, can one compute the proportion of unexplained range for a reference covariate(s) (e.g., patient age) and say something along the lines of "unobserved confounding would need to account for at least as much of the range in $Y$ as patient age to reach a sensitivity value of $\Lambda$"? Would such comparisons be valuable for judging the plausibility of different $\Lambda$ values?
* Along these lines, I don't think the authors discussed what reasonable values of $\Lambda$ might be in their cloud experiment. I think the exercise of reasoning about $\Lambda$ in the context of this example would be very helpful for readers and potential users of the method.


**Suggestions**
* Within the main paper, the authors assume a basic familiarity with marginal sensitivity models as known method/object (Tan, 2006) (e.g., "We propose CMSM as a new MSM" Ln 77). While they provide more information in the supplementary material, perhaps there is some minimal definition of a "marginal sensitivity model" that they can put in the main paper to provide a baseline context that they can build upon.
* To paint a fuller picture of the range of possible approaches to sensitivity analysis in the related work, the authors might consider adding a mention of works such as (Imbens (2003) and Veitch & Zaveri (2020)) which explicitly model the correlation due to unobserved confounding (for a binary treatment). Veitch & Zaveri gets around some criticism of existing methods that the authors discuss in Ln 246 by allowing for flexible, non-parametric models of the unobserved confounding relationship.

Imbens, G. W. (2003). Sensitivity to exogeneity assumptions in program evaluation. American Economic Review, 93(2), 126-132.

Veitch, V., & Zaveri, A. (2020). Sense and sensitivity analysis: Simple post-hoc analysis of bias due to unobserved confounding. Advances in Neural Information Processing Systems, 33, 10999-11009.

---

> ### Author Response · Authors · 2022-08-02
> **Thank you for your review 2/2**
>
> ### "Also, can one compute the proportion of unexplained range for a reference covariate(s) (e.g., patient age) and say something along the lines of "unobserved confounding would need to account for at least as much of the range in  as patient age to reach a sensitivity value of "? Would such comparisons be valuable for judging the plausibility of different  values?"
>
> In [this figure](https://i.imgur.com/QFuqLk3.png) we compare the same region with different covariates to identify an appropriate $\Lambda$. We fit one model on data from the Pacific (blue) and one model from the Pacific omitting $\omega_{500}$ from the covariates (orange). The shaded bounds in blue are the ignorance region for $\Lambda \to 1$ for the Pacific. We then find the $\Lambda$ that results in an ignorance interval around the Pacific omitting $\omega_{500}$ that covers the Pacific model prediction. From this, we can infer how the parameter $\Lambda$ relates to the inclusion of covariates in the model. We show that we need to set $\Lambda = 1.01$ to account for the fact that we are omitting $\omega_{500}$ from our list of covariates. Is this what you had in mind?
>
> ### "Along these lines, I don't think the authors discussed what reasonable values of  might be in their cloud experiment. I think the exercise of reasoning about  in the context of this example would be very helpful for readers and potential users of the method. In the climate experiment, what guidance is there for how to pick relevant values to consider? Could the authors add some kind of qualitative discussion/judgement here?"
>
> In the [this figure](https://i.imgur.com/H4X6ivE.png), we compare two regions with similar meteorology but different magnitudes of confounding influences to identify an appropriate $\lambda$. We fit one model on data from the Pacific region (blue) and one model on data from the Atlantic region (orange). The possible differing confounding factors include aerosol type, aerosol hygroscopicity, aerosol size, and others. We find the lambda that results in an ignorance interval around the Pacific model prediction that covers the Atlantic model prediction. From this, we can understand how well our current climate models (dashed: green, red, purple) recreate these trends and if their behavior is within the bounds which account for different confounding influences. Models outside of the shaded bounds, or regions of the model's predictions outside the shaded bounds, are likely not correctly emulating aerosol-cloud interactions, leading to substantial errors in their estimates of the effects.
>
> ### "How does the proposed method address possible positivity violations? The authors allude to this in the introduction but (it seems to me) it is never explicitly spelled out. It seems that the uncertainty quantification via bootstrapping is meant to capture uncertainty in, e.g., regions with poor overlap? I think some explicit statement about this (say, in Section 3.4) would help tie up any loose ends regarding the authors' claims."
>
> The statistical uncertainty quantified via bootstrapping ought to be high in regions of poor overlap. This hypothesis is put more explicitly by JMSG20 and JMGS21, who use different methods to quantify statistical uncertainty. We have added an explicit statement as suggested (lines 209-210).
>
> We offer [this figure](https://i.imgur.com/Xl1ExHp.png) showing an extended treatment axis for the results shown in Figure 4 as evidence. In the paper, we plot the 97th percentile of AOD (treatment) values in Figure 4, which lie roughly on (0.03, 0.4). The remaining 3% of observed treatment values lie roughly on (0.4, 3.0). Positivity is challenged here. As expected, the statistical (epistemic) uncertainty is very high where the overlap is weak.

---

> > ### Comment · Reviewer_UYQB · 2022-08-08
> > **Thanks for your responses**
> >
> > Thanks to the authors for their detailed responses to my questions. I remain very positive about this work and am maintaining my score.
> >
> > Regarding the responses about the real data experiment---I won't pretend to have a solid understanding of the scientific context for the climate application. On my initial read of Section 5.2 it felt like the main conclusion was that "as the strength of confounders increases (Λ > 1.0), the range of uncertainty in the treatment outcome increases", which is clearly how the method is intended to work, and there was less of a story regarding bringing clarity to the climate application itself. However, upon another read and in light of some of the rebuttal answers, it seems that the authors do in fact interpret $\Lambda$ in the context of the application. Overall, I think this section might benefit from some more explanation given more space in a camera ready version of the paper.

---

> > > ### Author Response · Authors · 2022-08-09
> > > **Thank you**
> > >
> > > Thank you again for your insightful review. Your suggestion to expand upon the interpretation and discussion of the climate results will clearly improve the paper and we will certainly do so if an extra page is granted.

---

> ### Author Response · Authors · 2022-08-02
> **Thank you for your review 1/2**
>
> Thank you very much for your detailed feedback. We are sincerely flattered by your positive comments. We address your questions and concerns below.
>
> TL;DR — *We offer a minimal definition of a marginal sensitivity model. We propose to provide a broader literature review in the appendix. We provide additional insights into understanding the $\Lambda$ parameter using the climate data. We elaborate on the connections between statistical uncertainty and violations of positivity.*
>
> ### "Within the main paper, the authors assume a basic familiarity with marginal sensitivity models as known method/object (Tan, 2006) (e.g., "We propose CMSM as a new MSM" Ln 77). While they provide more information in the supplementary material, perhaps there is some minimal definition of a "marginal sensitivity model" that they can put in the main paper to provide a baseline context that they can build upon."
>
> The term "marginal sensitivity model" is introduced by ZSB19. For ZSB19, a sensitivity model is a user hypothesized complete propensity score function. Analogously, a continuous treatment sensitivity model in our setting would be a user hypothesized propensity density. The inverse of the hypothesized propensity density is:
>
> $\frac{1}{p(\mathrm{t} \mid \mathrm{y}, \mathbf{x})} = \frac{1}{\Lambda p(\mathrm{t} \mid \mathbf{x})} + w(\mathrm{y}, \mathbf{x}) \left( \frac{\Lambda}{p(\mathrm{t} \mid \mathbf{x})}-\frac{1}{\Lambda p(\mathrm{t} \mid \mathbf{x})}\right).$
>
> For ZSB19, they define their marginal sensitivity model as the collection of user hypothesized complete propensity score functions that satisfy the bound on the odds ratio. Analogously, we could define the CMSM as
>
> $\mathcal{P}(\Lambda) = \{ p(\mathrm{t} \mid \mathrm{y}, \mathbf{x}): \frac{1}{\Lambda} \leq \frac{p(\mathrm{t} \mid \mathbf{x})}{p(\mathrm{t} \mid \mathrm{y}, \mathbf{x})} \leq \Lambda, \forall \mathbf{x} \in \mathcal{X}, \forall \mathrm{y} \in \mathbb{R} \}$.
>
> The brackets are not compiling here, but we have added the definition to the manuscript (lines 102-105).
>
> Your comment does bring up an interesting question: why marginal? Again, ZSB19 introduces the terminology. They attribute MSM to Tan06, but it is not used by Tan06. Further, they do not elaborate on why marginal. We speculate that this could either be because the models are rooted in the generally unidentifiable marginal distribution of potential outcomes $P(Y_t \mid X=x)$, or because you marginalize over the treatment with respect to the hypothesized inverse propensity score. Perhaps both. We will follow up with ZSB once it is clear that it will not jeopardize anonymity.
>
> ### "To paint a fuller picture of the range of possible approaches to sensitivity analysis in the related work, the authors might consider adding a mention of works such as (Imbens (2003) and Veitch & Zaveri (2020)) which explicitly model the correlation due to unobserved confounding (for a binary treatment). Veitch & Zaveri gets around some criticism of existing methods that the authors discuss in Ln 246 by allowing for flexible, non-parametric models of the unobserved confounding relationship."
>
> Thank you for sharing these works. We propose to add a thorough discussion in the appendix to include this related literature. Reviewers eRS3 and zMDP have also suggested literature on the discrete treatment regime, so it is clear that this change will strengthen the paper.
>
> ### "The main possible difficulty with the proposed method that came to mind is how to select values of the sensitivity parameter. As opposed to the odds ratio (which, as the authors note, is generally interpretable to practitioners), the density ratio is somewhat difficult to make judgements about directly. The authors provide an alternative characterization in terms of the "proportion of unexplained range in ", but even this is (to my knowledge) not a commonly considered statistic. Can this be related in any way to, e.g., the  (i.,e., the fraction of variance unexplained)?"
>
> Good question. It seems non-trivial to make the direct connection between Lambda and the fraction of unexplained variance. We think this is better left as a future contribution if it turns out to be possible. Indeed, we propose the proportion of unexplained range as an intermediate heuristic reflecting the fraction of unexplained variance we would attribute to hidden confounding under an assumed $\Lambda$. We are also exploring methods using quantiles of the conditional distribution of the outcome.

---

### Official Review · Reviewer_eRS3 · 2022-07-19

**Rating:** 8
**Confidence:** 4
**Soundness:** 3 good
**Presentation:** 4 excellent
**Contribution:** 4 excellent

**Summary:**

The paper tackles a very interesting an important question, that of bounding treatment effects when ignorability is violated and the treatment and covariates are continuous. They do this by proposing an extension of the marginal sensitivity model (MSM) to the so called continuous marginal sensitivity model (CMSM). In the CMSM, a density ratio is used to quantify the belief of how much ignorability is violated, in contrast to the odds ratio used in MSM.

Following this the authors five a semi-parametric estimator for the bounds and an algorithm to compute them. The claims are then validated though synthetic and real world experiments.

**Questions:**

* Are there any baselines that the authors could have compare their methods to? For instance even something line shoehorning a continuous MSM into the binary setting (for a single dimensional continuous X of course)?
* While the amount of compute used is reasonably small, I couldn't find any information on how long it took to get the bounds? Do you some estimate of this?


**Limitations:**

The  potential negative societal impact is discussed but the limitations could be described better.

**Strengths And Weaknesses:**

**Strengths**.
* The paper is well written and easy to follow in my opinion. The flow is natural and it is clear how each part fits into the bigger picture while reading the paper.
* The paper evaluates their method on a real world datasets in collaboration with the domain experts. The ideal application of causal inference is always said to be in alliance with domain experts, and using that as one of the ways to evaluate a proposed method speaks in favour of the method and the paper.
* The experiments seem to justify the claims made in the paper.
* The appendix provides a thorough description of the background on MSM and how the proposed method relates to it.

**Weaknesses**.
* The limitations of the method should be discussed to give a clearer picture of the method. One limitation is could be the need for a grid search.

**Other related work**. Considering that the paper deals with bounding causal effects, the following papers could be mentioned in the background work. While they are not directly doing sensitivity analysis per se, I think they are relevant to getting a context of the related work since they also tackle the problem of bounding treatment effects in the presence of confounding.

*Continuous setting*:
Kilbertus, N., Kusner, M.J. and Silva, R., 2020. A class of algorithms for general instrumental variable models. Advances in Neural Information Processing Systems, 33, pp.20108-20119.

Hu, Y., Wu, Y., Zhang, L. and Wu, X., 2021, May. A generative adversarial framework for bounding confounded causal effects. In Proceedings of the AAAI Conference on Artificial Intelligence (Vol. 35, No. 13, pp. 12104-12112).

Padh, K., Zeitler, J., Watson, D., Kusner, M., Silva, R. and Kilbertus, N., 2022. Stochastic Causal Programming for Bounding Treatment Effects. arXiv preprint arXiv:2202.10806.

*Discrete setting*:
Duarte, Guilherme, Noam Finkelstein, Dean Knox, Jonathan Mummolo, and Ilya Shpitser. 2021a. “An Automated Approach to Causal Inference in Discrete Settings.” arXiv [stat.ME]. arXiv. http://arxiv.org/abs/2109.13471.

Duarte, G., Finkelstein, N., Knox, D., Mummolo, J. and Shpitser, I., 2021. An automated approach to causal inference in discrete settings. arXiv preprint arXiv:2109.13471.

Zhang, Junzhe, Jin Tian, and Elias Bareinboim. 2021. “Partial Counterfactual Identification from Observational and Experimental Data.” arXiv [cs.AI]. arXiv. https://www.causalai.net/r78.pdf.

---

> ### Author Response · Authors · 2022-08-02
> **Thank you for your review**
>
> Thank you very much for your detailed feedback. We are pleased you found the paper well written and appreciate our efforts to evaluate the method in a real-world context. We address your questions and concerns below.
>
> TL;DR — *We refer you to the appendix for alternatives to the grid-search algorithm. We comment on the surprising efficacy and speed of the grid search. We have updated the main paper to include the continuous treatment works and will provide a broader literature review in the appendix. We need clarification on the experiment you would expect.*
>
> ### "The limitations of the method should be discussed to give a clearer picture of the method. One limitation is could be the need for a grid search."
>
> We have tried to be transparent about the limitations of our method by clearly stating all assumptions, commenting on the challenges with interpreting $\Lambda$, and highlighting the current gaps in theory that delay the arrival of causal conclusions from observational climate data. We appreciate that there may always be further limitations not yet considered.
>
> Regarding the need for a grid search, we refer you to appendices D.0.1 and F, where we give line-search and gradient descent alternatives. Initial unreported results have shown that both options increase algorithmic complexity without improving the compute time or bound fidelity. The "grid search is all you need" phenomenon is probably due to modern GPU architectures. It is parallelizable, and the bounds for a batch of data can be computed in 3ms using an NVIDIA 1080 Ti and consumer CPU. Analyzing different algorithms could be interesting for future work but is beyond the scope of this paper since the grid search is surprisingly effective and a solution to the problem.
>
> ### "Considering that the paper deals with bounding causal effects, the following papers could be mentioned in the background work."
>
> Thank you for sharing these works. The instrumental variable approaches of KKS20, HWZW, and PZWKSK22 are very relevant and escaped our review. We have updated the related works section to include these (line 259). Reviewers UYQB and zMDP have also shared additional works in the discrete treatment setting, and we propose to add these to a detailed lineage in the appendix due to space constraints in the main paper.
>
> ### "Are there any baselines that the authors could have compare their methods to? For instance even something line shoehorning a continuous MSM into the binary setting (for a single dimensional continuous X of course)?"
>
> We have to ask for clarification here. By "single dimensional continuous X," we assume you are looking for a synthetic experiment. Then, would we use the density ratio (probability ratio) instead of the odds ratio for a binary treatment setup?
>
> ### "While the amount of compute used is reasonably small, I couldn't find any information on how long it took to get the bounds? Do you some estimate of this?"
>
> The following estimates are for a consumer intel CPU, 16GB of RAM, and an NVIDIA GeForce GTX 1080 Ti GPU. For both the neural network architecture and transformer architecture, a forward pass and bound estimation on a batch of data is completed on the order of miliseconds.

---

> > ### Comment · Reviewer_eRS3 · 2022-08-09
> > **Thank you for the answers**
> >
> > Thank you very much for the detailed response. Again, it was a fun paper to read.
> >
> > For the experiment, what I meant was something like the comparison done in [1] (Page 8, first column last paragraph, dashed yellow lines in the experiments was what I was referring to). In any case it was a minor point which I asked out of curiosity, and not really expected. I am already satisfied with all the evaluations presented in the paper (hence my high score). After reading all the reviews and the authors' responses to them, I am happy to increase my score for the paper.
> >
> > [1] Marmarelis, M. G., Steeg, G. V., & Galstyan, A. (2022). Bounding the Effects of Continuous Treatments for Hidden Confounders. arXiv preprint arXiv:2204.11206.

---

> > > ### Author Response · Authors · 2022-08-09
> > > **Thank you**
> > >
> > > Thank you again for the review and we appreciate the score increase as it ought to help increase the visibility of our work. Thank you for your clarification regarding the experiment. We will aim to include this in the camera ready should this paper be accepted.

---

### Author Response · Authors · 2022-08-02
**To all reviewers**

A sincere thank you to all reviewers for being so generous with their time and attention in reviewing our work. We are greatly encouraged that you have each given scores indicating that our work is worthy of acceptance to NeurIPS. There do not seem to be any major concerns with our theoretical or empirical results. We have updated our manuscript according to the suggestions you have made and plan to extend the literature review in the appendix.

We are also encouraged by your reaction to our empirical evaluation using climate data. In order to verify recent improvements in climate modeling, we must first improve how we validate and establish trends from our observational record. Unfortunately, confounding effects such as the swelling of aerosol in humid environments, makes it difficult to confidently establish baseline trends that we can compare to model output. The methodology and resulting model within allow climate modelers and observationalists to established bounds of possible effects while accounting for these confounding influences. This not only allows us to understand which models can recreate the observed trends, but how well they recreate the trends within distinct environmental regimes.

---

### Meta-Review · Area_Chair_4Goa · 2022-08-27

**Recommendation:** Accept
**Confidence:** Certain

**Metareview:**

This paper extends the marginal sensitivity model to continuous treatments. Given the developments in the discrete treatment setting, none of the parts of the paper are surprising. Further, there are several simultaneous related works that carry out a generalization to continuous treatments. That being said, the work is sound and a polished contribution.

**Award:**

No

---

### Decision · Program_Chairs · 2022-09-14

Accept